# PaperQA: Retrieval-Augmented Generative Agent for Scientific Research

## Abstract

Large Language Models (LLMs) generalize well across language tasks, but suffer from hallucinations and uninterpretability, making it difficult to assess their accuracy without ground-truth. Retrieval-Augmented Generation (RAG) models have been proposed to reduce hallucinations and provide provenance for how an answer was generated. Applying such models to the scientific literature may enable large-scale, systematic processing of scientific knowledge. We present PaperQA, a RAG agent for answering questions over the scientific literature. PaperQA is an agent that performs information retrieval across full-text scientific articles, assesses the relevance of sources and passages, and uses RAG to provide answers. Viewing this agent as a question-answering model, we find it exceeds performance of existing LLMs and LLM agents on current science QA benchmarks. To push the field closer to how humans perform research on scientific literature, we also introduce LitQA, a more complex benchmark that requires retrieval and synthesis of information from full-text scientific papers across the literature. Finally, we demonstrate PaperQA's matches expert human researchers on LitQA.

## 1 Introduction

The rate of papers published yearly grows at an exponential rate, with over 5 million academic articles published in 2022 (Curcic, 2023), and over 200 million articles in total (Fire & Guestrin, 2019). The difficulty of navigating this extensive literature means significant scientific findings have gone unnoticed for extended periods (Fortunato et al., 2018; Stent, 1972; Garfield, 1980). Work in the last 10 years has sought to make the space of literature more manageable for scientists, with the introduction of keyword search systems (Google Scholar; Kinney et al., 2023), vector similarity embeddings (Cohan et al., 2020; Beltagy et al., 2019; Bojanowski et al., 2017; Singh et al., 2022) and recommender systems (Resnick & Varian, 1997; Adomavicius & Tuzhilin, 2005). The process of scientific discovery from literature is still, however, highly manual.

The use of Large Language Models (LLMs) to answer scientific questions is increasingly seen in academia and research-heavy professions such as medicine (Singhal et al., 2023; Williams et al., 2023). While LLMs can produce answers faster and encompass a broader, deeper scope than manual searching, there is a high risk of hallucination in responses, which can lead to potentially dangerous outcomes (Hiesinger et al., 2023; Meskó & Topol, 2023). Incorrect information can be more damaging than no information at all, as the time to verify veracity can take just as long as retrieving it from research papers in the first place (Williamson, 2016). Reliance on pre-trained LLMs also prevents the discovery of new information published after a training cutoff date. Given the rapidly moving pace of science, this can lead to misconceptions persisting.

Retrieval-Augmented Generation (RAG) models are a potential solution to these limitations (Lewis et al., 2020b). RAG models retrieve text from a corpus, using methods such as vector embedding search or keyword search, and add the retrieved passage to the context window of the LLM. RAG usage can reduce hallucinations in conversations and improve LLM performance on QA tasks (Hiesinger et al., 2023; Shuster et al., 2021). Nevertheless, standard RAG models follow a fixed, linear flow, which can be restrictive for addressing the diverse range of questions scientists encounter.

In this work, we eliminate these limitations by breaking RAG into modular pieces, allowing an agent LLM to dynamically adjust and iteratively perform steps in response to the specific demands of each question, ensuring more precise and relevant answers. We call this PaperQA, an agent-based

RAG system for scientific question answering. PaperQA has three fundamental components: finding papers relevant to the given question, gathering text from those papers, and generating an answer with references.

We evaluate PaperQA on several standard multiple-choice datasets for evaluating LLMs, which assess the model's ability to answer questions with existing knowledge, but do not test the agents' ability to retrieve information. We modified PubMedQAJin et al. (2019) to remove the provided context (so it is closed-book) and found PaperQA beats GPT-4 by 30 points (57.9% to 86.3%). PubMedQA is only built on abstracts though, and so we construct a more difficult dataset that requires synthesizing information from one or multiple full-text research papers. We thus introduce a new dataset, LitQA, composed from recent literature, in order to test PaperQA's ability to retrieve information outside of the underlying LLM's pre-training data. PaperQA outperforms all models tested and commercial tools, and is comparable to human experts on LitQA on performance and time, but is significantly cheaper in terms of costs. Furthermore, PaperQA is competitive to state-of-the-art commercial tools for scientific question answering. Finally, we find that PaperQA exhibits a better knowledge boundary than competing tools, answering questions incorrectly at a lower rate, and instead, answering that it is unsure.

## 2 RELATED WORKS

**LLMs for Natural Sciences**   Large Language Models (LLMs) have seen a surge in popularity and accessibility, leading to impressive applications across various domains (Vaswani et al., 2017; Brown et al., 2020; Bommasani et al., 2021; Chowdhery et al., 2022; Driess et al., 2023; OpenAI, 2023). LLMs have been used effectively for text-based scientific tasks, such as extracting chemical reaction procedures (Bai et al., 2022) and entity extraction from biological documents (Tamari et al., 2021). LLMs trained on biomedical (Gu et al., 2021; Lewis et al., 2020a; Luo et al., 2022; Lee et al., 2020; Shin et al., 2020) or more generalized scientific literature (Beltagy et al., 2019; Taylor et al., 2022) have demonstrated impressive performance on current scientific QA benchmarks.

State-of-the-art pre-trained LLMs exhibit proficiency in complex tasks such as searching for chemical compounds (OpenAI, 2023) and designing new catalysts (Ramos et al., 2023). Still, LLMs frequently fall short in many scientific domains due to outdated knowledge (Fan et al., 2022) and reasoning problems (Shinn et al., 2023; Wang et al., 2022). Prompt engineering techniques can enhance LLMs' reasoning abilities (Wei et al., 2022; Yao et al., 2023), though some scientific tasks requiring real-time calculations and up-to-date information remain difficult.

**Agents**   Another technique is to integrate external tools into LLMs, creating agent systems, such as MRKL Karpas et al. (2022). These LLM-agent systems leverage the reasoning power of the base-LLM and the use of pre-defined external tools to *act* in order to complete a given prompt Schick et al. (2023); Shen et al. (2023). In such a system, the agent iteratively decides on the best tools to use in order to address the given task, correcting previous behavior when necessary, until the task is complete. This setup can also integrate reasoning steps, as with the ReAct framework and its derivatives (Yao et al., 2022; Yang et al., 2023b), or may incorporate multi-agent systems (Talebirad & Nadiri, 2023; Wang et al., 2023; Significant-gravitas, 2023). Our method extends previous work by integrating an agent-based RAG framework in order to correctly and dependably answer scientific questions.

**Evaluating LLM Scientists**   Assessing the scientific capabilities of LLMs often relies on QA benchmarks, such as general science benchmarks (Kembhavi et al., 2017; Hendrycks et al., 2020), or those specializing in medicine (Jin et al., 2019), biomedical science (Sung et al., 2021) or chemistry (Guo et al., 2023; Wu et al., 2017). In contrast, open-ended tasks, such as conducting chemical synthesis planning (Bran et al., 2023) or offering healthcare support (Dash et al., 2023), necessitate manual evaluation to effectively measure an LLM's capabilities. We discuss the limitation of these datasets and introduce a new QA dataset for retrieval-based science.

**Retrieval-Augmented LLMs**   LLMs integrated with retrieval systems are broadly referred to as Retrieval-Augmented Generation (RAG) models (Lewis et al., 2020b), with the key components being a database of documents, a query system for retrieving documents, and a pipeline to add retrieved documents to a model's context. RAG models have proved effective at biomedical and

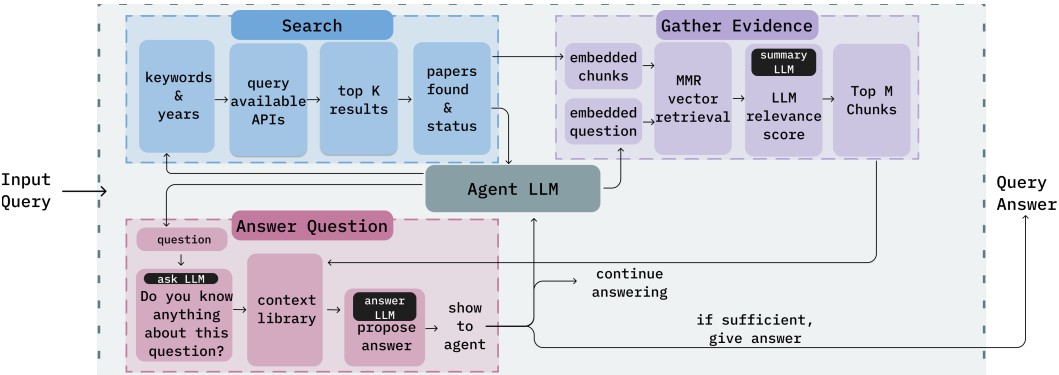

Figure 1: **PaperQA Workflow Diagram.** PaperQA is an agent that transforms a scientific question into an answer with cited sources. The agent utilizes three tools – search, gather evidence, and answer question. The tools enable it to find and parse relevant full-text research papers, identify specific sections in the paper that help answer the question, summarize those section with the context of the question (called evidence), and then generate an answer based on the evidence. It is an agent, so that the LLM orchestrating the tools can adjust the input to paper searches, gather evidence with different phrases, and assess if an answer is complete.

clinical QA tasks (Soong et al., 2023; Hiesinger et al., 2023). While RAG systems are typically pipelines with a fixed number and order of steps, they can be composed of multi-hop searches (Khattab et al., 2022), or use Agents (Significant-gravitas, 2023), where an LLM is used to decide when to use retrieval to augment an answer (Schick et al., 2023). RAG performance can be further improved with *a priori* prompting, where an LLM is prompted to consider using latent information instead of searching, or *a posteriori* prompting, where an LLM is asked to consider the accuracy of a retrieved result and possibly provide an alternative answer (Ren et al., 2023). Active RAG (Jiang et al., 2023) regenerates sentences using retrieval if they contain low-probability tokens.

**Retrieval Methods** RAG models connected to databases retrieve documents using fixed representations, such as Bag-of-Words or BM25 (Izacard & Grave, 2020; Doostmohammadi et al., 2023)), pre-trained embeddings (Devlin et al., 2018; Izacard & Grave, 2020), or trainable encoders (Lewis et al., 2020b; Shi et al., 2023), which are trained offline (Shi et al., 2023) or online (Guu et al., 2020). For models using open-ended sources such as the internet, phrase or keyword searches are used for retrieval (Menick et al., 2022; Significant-gravitas, 2023). Retrieved documents are typically passed to the model as input tokens, but some models separately process retrieved information with cross-attention (Borgeaud et al., 2022; Doostmohammadi et al., 2023). Learning to rank and filter the retrieved documents (Lyu et al., 2023) further improves the performance of RAG models. In our work, we evaluate Retrieval-Augmented Models that use keyword search systems and vector embedding retrieval. In particular, we focus on benchmarking against humans and uncover under what conditions LLMs can match human performance in information retrieval.

## 3 METHOD

### 3.1 PAPERQA

The PaperQA system is an agent shown in Figure 1. The fundamental operations of PaperQA are to find relevant papers from online databases, such as Arxiv[1] and Pubmed[2]; gather text from these papers; and synthesize information into a final answer. PaperQA's tools are composed of both external APIs and LLMs, described in detail below. Compared to a standard RAG, we make four key innovations. First, we decompose parts of a RAG and provide them as tools to an agent. This allows for some operations, such as search, to be performed multiple times, with different keywords, if the evidence collected is insufficient. Second, we use a *map* summarization step to

---

[1]https://arxiv.org/
[2]https://pubmed.ncbi.nlm.nih.gov/

gather evidence from multiple sources followed by a *reduce* step to answer. The map-reduce step increases the amount of sources that can be considered, and provides a scratchpad (Wei et al., 2022), where the LLM can give intermediate *evidence* before formulating the final answer. Third, we utilise the LLM's ability to reason over text and provide numerical relevance scores for each chunk of text to the query question. In addition to the commonly used vector-embedding distances, we use these LLM-generated relevancy scores, adding another layer of retrieval. Finally, we make use of *a priori* and *a posteriori* prompting, tapping into the latent knowledge in LLMs.

## 3.2 TOOLS

PaperQA has three tools: `search` that finds relevant papers, `gather_evidence` that collects most relevant chunks of papers relative to the query into a context library, and `answer_question` which proposes an answer based on the accumulated contexts. These tools make use of three independent LLM instances, a `summary LLM`, an `ask LLM`, and an `answer LLM`, which ultimately take a string as input and output a status string while accumulating evidence to answer the question. The system prompts and the tools' descriptions appear in Appendices A.2 and A.3 respectively. The `agent LLM` is initialised with the following prompt:

```
Answer question: question. Search for papers, gather evidence, and answer. If you do not
have enough evidence, you can search for more papers (preferred) or gather more evidence
with a different phrase. You may rephrase or break-up the question in those steps. Once
you have five or more pieces of evidence from multiple sources, or you have tried many
times, call answer_question tool. You may reject the answer and try again if it is
incomplete.
```

**search** The agent gives this tool keywords and (optionally) a year range. This queries a scientific literature search engine. Retrieved papers are added to a local bibliography for use by `gather_evidence`. They are added by creating overlapping 4,000 character chunks which are embedded with the `text-embedding-ada-002` model (Neelakantan et al., 2022) and inserted into a vector database (Johnson et al., 2019). There is a failure rate associated with the performance of search engines, accessing papers, and parsing of PDFs. We explore this in Appendix C. This tool is always executed once with the full-text question before initialising the agent.

**gather_evidence** This tool is given a question as input by the agent. This question is vector-embedded using the OpenAI `text-embedding-ada-002` (Neelakantan et al., 2022), and relevant chunks based on vector search are returned from the vector database created in the `search` tool. Chunks are retrieved with *maximal marginal relevance* search (Carbonell & Goldstein, 1998) to increase the diversity of returned texts. Each retrieved chunk is fed to a `summary LLM` with the following prompt:

```
Summarize the text below to help answer a question. Do not directly answer the question,
instead summarize to give evidence to help answer the question. Reply 'Not applicable' if
text is irrelevant. Use summary_length. At the end of your response, provide a score from
1-10 on a newline indicating relevance to question. Do not explain your score.

chunk

Excerpt from citation
Question: question
Relevant Information Summary:
```

The returned chunks are then sorted by score and the top-k chunks are collected in a *context library*. The tool returns the top-1 chunk to the agent, so that the agent can respond to the top chunk.

The `gather_evidence` tool is a map step that is always present in RAG systems. It provides an opportunity to reject irrelevant context and can be done concurrently to minimize the compute time. It is especially helpful for PDF parsing errors, which can be mitigated by having this step that can summarize garbled text with the question providing context to guide the summary.

**answer_question** We first use the `ask LLM` to provide any useful information from the pre-trained LLM that might help with answering the original question (details are given in Appendix A.4),

similar to the to the priori judgement explored in (Ren et al., 2023). The output of the `ask` LLM is added to the chunks from the context library, and the final prompt to the `answer` LLM is as follows:

```
Write an answer (answer_length) for the question below based on the provided context. If
the context provides insufficient information, reply "I cannot answer". For each part of
your answer, indicate which sources most support it via valid citation markers at the end
of sentences, like (Example2012). Answer in an unbiased, comprehensive, and scholarly
tone. If the question is subjective, provide an opinionated answer in the concluding 1-2
sentences.

context

Extra background information: ask LLM

Question: question

Answer:
```

The question in this prompt is the original question fed to the higher-level PaperQA agent, and the context comes from the collection of relevant chunks within the context library. The response from the the tool is returned to the agent, which may reject or accept the answer based as given in the agent's initialisation prompt given above.

## 4  THE LITQA DATASET

Existing benchmarks for scientific question-answering evaluate the ability of AI systems to present widely-known or widely-available information, or to answer questions given a specific context. These benchmarks are thus insufficient for evaluating an agent's ability to answer questions based on retrieved information. We thus introduce the LitQA dataset for PaperQA evaluation. The questions in LitQA are sourced from recent literature (after September 2021), so that we expect that they will not have been including in training for most language models, and are designed to be difficult or impossible to answer without retrieving the relevant paper. LitQA thus evaluates the ability both to retrieve necessary information and to present an accurate answer based on that information.

**Dataset description** The LitQA dataset consists of 50 multiple-choice questions, assembled by experts. All questions come from the biomedical domain. It has 5 Yes/No questions, 6 questions with 3 possible answers, 23 questions with 4 possible answers, 10 questions with 5 possible answers, 4 questions with 6 possible answers and finally 2 questions with 7 possible answers. We show examples of the questions in Table 1.

**Data collection** All questions were written and reviewed by researchers in natural and biomedical sciences. Questions were assembled by first selecting a paper published after the September 2021 cutoff and then devising a question from that paper that both refers to a novel finding in the paper and that is not presented in the paper abstract. We do not expect these findings to be presented in other papers or to be available in sources preceding the cutoff. For every question, distractor answers were also created, either by the question writer alone, or by using an LLM to provide a plausible answer to the given question. We take special care to only collect questions from papers published *after* the GPT-3.5/4 cutoff date in September 2021. This ensures the questions cannot be reliably answered using the latent knowledge of GPT-4. Each question was then independently reviewed by at least one other co-author.

**Human performance** We recruited five biomedical researchers with an undergraduate degree or higher to solve LitQA. They were given access to the internet and given three minutes per question (2.5 hours in total) to answer all questions, specifying the paper they used to choose the corresponding answer. Additionally, they were instructed to answer *unsure* if they cannot find the answer to a given question.

## 5  EXPERIMENTS

In this section we measure the performance of PaperQA on LitQA and other datasets, and compare to existing commercial products. We ablate all components of PaperQA on the LitQA dataset,

Table 1: **LitQA Dataset Examples.** Three example questions from the dataset with correct answers bolded. The difficulty annotated comes from our observations in experiments given in Section 5.

| ID | Question | Answers | Difficulty |
|----|----------|---------|------------|
| 7 | Has anyone performed a base editing screen against splice sites in CD33 before? | **Yes** 
 No | Easy |
| 32 | How diffuse are the laminar patterns of the axonal terminations of lower Layer 5/Layer 6 intratelencephalic neurons compared to Layer 2-4 intratelencephalic neurons in mouse cortex? | **More diffuse** 
 About the same 
 Less diffuse | Medium |
| 21 | Which of these glycoRNAs does NOT show an increase in M0 macrophages upon stimulation with LPS: U1, U35a, Y5 or U8? | **U8** 
 U1 
 U35a 
 Y5 | Hard |

showing their importance. Finally, we investigate the ability to retrieve relevant papers and the rate of hallucinations.

## 5.1 EXPERIMENTAL DETAILS

All of the experiments were implemented within LangChain's agent framework (Chase, 2022). As described above, we use four different LLM instances with the following settings unless stated otherwise: `agent LLM`: GPT-4 OpenAIFunctions Agent with temperature $\tau_{\text{agent}} = 0.5$; `summary LLM`: GPT-3.5 always with temperature $\tau_{\text{sum}} = 0.2$; `answer LLM`: GPT-4 with temperature $\tau_{\text{ans}} = 0.5$; `ask LLM`: GPT-4 with temperature $\tau_{\text{ask}} = 0.5$. For the search engine, we use Google Scholar, where we collect the top-5 papers that are accessible. The implementation details are explained in an experiment in Appendix C. We access papers through publicly available APIs, such as Arxiv, PMC, OpenAccess, PubMed, and our own local database of papers. Lastly, we set the number of sources to consider at each round of `gather_evidence` to 20, and the count of evidence context to include in the final answer within `answer_question` to 8.

## 5.2 RESULTS

Here we thoroughly evaluate PaperQA. We compare it against commercially available tools and human performance on LitQA. Next, we ablate the components of PaperQA and look into hallucinated citations. Finally, we evaluate PaperQA on several standard QA benchmarks.

**Scientific Question Answering** First, we compare PaperQA on LitQA with two pre-trained LLMs, AutoGPT and several commercially available tools for scientific research – Elicit, Scite_ and Perplexity – in Table 2. All commercial tools are specifically tailored to answering questions by retrieving scientific literature. We give them the same prompt as to PaperQA. From Table 2 we see that PaperQA outperforms all competing models and products, and is on par with that of human experts with access to the internet. Furthermore, we see the lowest rate of incorrectly answered questions out of all tools, which rivals that of humans. This emphasizes that PaperQA is better calibrated to express uncertainty when it actually is uncertain. Surprisingly, GPT-4 and Claude-2 perform better than random although the questions are from papers after their training cut-off date, suggesting they have latent knowledge, leading to useful bias towards answers that are more plausible.

PaperQA averaged 4,500 tokens (prompt + completion) for the more expensive LLMs (`agent LLM`, `answer LLM`, `ask LLM`) and 24,000 tokens for the cheaper, high-throughput LLM (`summary LLM`). Based on commercial pricing as of September 2023, that gives a cost per question of $0.18 using the stated GPT-4 and GPT-3.5-turbo models. It took PaperQA on average about 2.4 hours to answer all questions, which is also on par with humans who were given 2.5 hours. A single instance of PaperQA would thus cost $3.75 per hour, which is a fraction of an average hourly wage of a desk researcher. We exclude other negligible operating costs, such as search engine APIs, or electricity.

**How does PaperQA compare to expert humans?** PaperQA shows similar results to those of the expert humans who answered the questions. To quantify this, we calculate the categorical cor-

Table 2: **Evaluation on LitQA**. We compare PaperQA with other LLMs, the AutoGPT agent, and commercial products that use RAG. AutoGPT was run with GPT-4, where other implementation details are given in the Appendix B. Elicit.AI was run on default settings, Perplexity was run in academic mode, Perplexity Co-pilot was run on default settings (perplexity model, "all sources"), and "Assistant by Scite_" was run on default settings. Each question was run on a new context (thread) and all commercial products were evaluated on September 27, 2023. We report averages over a different number of runs for each.

| Model | Samples | Response | | | Score | |
|---|---|---|---|---|---|---|
| | | Correct | Incorrect | Unsure | Accuracy ($\frac{Correct}{All}$) | Precision ($\frac{Correct}{Sure}$) |
| Random | 100 | 10.2 | 29.5 | 10.3 | 20.4% | 25.7% |
| Human | 5 | 33.4 | 4.6 | 12.0 | 66.8% | **87.9%** |
| Claude-2 | 3 | 20.3 | 26.3 | 3.3 | 40.6% | 43.6% |
| GPT-4 | 3 | 16.7 | 16.3 | 17.0 | 33.4% | 50.6% |
| AutoGPT | 3 | 20.7 | 7.3 | 22.0 | 41.4% | 73.9% |
| Elicit | 1 | 12.0 | 16.0 | 22.0 | 24.0% | 42.9% |
| Scite_ | 1 | 12.0 | 21.0 | 17.0 | 24.0% | 36.4% |
| Perplexity | 1 | 9.0 | 10.0 | 31.0 | 18.0% | 47.4% |
| Perplexity (Co-pilot) | 1 | 29.0 | 10.0 | 12.0 | 58.0% | 74.4% |
| PaperQA | 4 | 34.8 | 4.8 | 10.5 | **69.5%** | **87.9%** |

relation (Cramer's $V$) of the responses for each human-human and human-PaperQA pair. Average human-human $V$ is $0.66 \pm 0.03$, whereas average human-PaperQA $V$ is $0.67 \pm 0.02$ (mean $\pm$ stderr), indicating that PaperQA is, on average, as correlated with human respondents as the human respondents are with each other, implying no discernable difference in responses. To compare, the average $V$ between humans and Perplexity was $0.630 \pm 0.05$.

Table 3: **Ablation of LLM Types on LitQA (left).** We compare using different LLMs for the `answer LLM` and `summary LLM`, averaged over 2 runs. **Ablation of PaperQA Components on LitQA (right).** *No summary LLM* setting only implements vector retrieval without any summarizaton and relevance scoring. *Single citation* only uses the best scoring chunk from the context library. *No ask LLM* ignores including the LLM's latent knowledge within the context of the `answer LLM`. *No search* setting starts with all papers from LitQA already collected and runs `gather_evidence` once. *Vanilla RAG* setting runs the `search` tool thrice, followed by a single call to `gather_evidence` and `answer_question`. *Semantic Scholar* uses Semantic Scholar instead of Google Scholar. *No MC options* provides no multiple-choice answers as options within the question prompt. All ablations were done with a single run.

| Model | | Correct | Incorrect | Unsure |
|---|---|---|---|---|
| Answer | Summary | | | |
| Claude-2 | GPT-3.5 | 32.5 | 7.0 | 10.5 |
| Claude-2 | GPT-4 | 26.5 | 9.0 | 14.5 |
| GPT-4 | GPT-3.5 | 33.5 | 5.0 | 11.5 |
| GPT-4 | GPT-4 | 31.5 | 7.0 | 11.5 |

| Ablation | Correct | Incorrect | Unsure |
|---|---|---|---|
| PaperQA | 33.5 | 5.0 | 11.5 |
| No summary LLM | 29.0 | 10.0 | 11.0 |
| Single citation | 27.0 | 10.0 | 13.0 |
| No ask LLM | 23.0 | 14.0 | 13.0 |
| No search | 23.0 | 7.0 | 20.0 |
| Vanilla RAG | 22.0 | 6.0 | 22.0 |
| Semantic Scholar | 21.0 | 4.0 | 25.0 |
| No MC options | 18.0 | 6.0 | 26.0 |

**Ablating PaperQA** We report performance on LitQA when toggling different parts and LLMs of PaperQA in Table 3. Using GPT-4 as the `answer LLM` slightly outperforms Claude-2. When we look at the different components of PaperQA, we observe a major drop in performance when not including multiple-choice options as answers (*no MC options*) and using *Semantic Scholar* instead of Google Scholar. The former we explain with the fact that closed-form questions are easier than open-form ones, and the model can use keywords derived from the possible answers to search.

Table 4: **Comparison of Hallucination in Citations.** We compare PaperQA's hallucination rates of citations with three pre-trained LLMs. *Hallucinated* refers to *references* that are either nonexistent, partially inaccurate, or have content that does not match the claim. Scores are calculated for each LLM based on N number of citations provided by the model.

| LLM | Valid (%) | Hallucinated (%) | | | N |
| --- | --- | --- | --- | --- | --- |
| | | Full Hallucination | Citation Inaccuracy | Context Irrelevance | |
| GPT-3.5 | 52.50% | 33.75% | 12.50% | 1.25% | 80 |
| GPT-4 | 60.78% | 29.41% | 3.92% | 5.88% | 51 |
| Claude-2 | 39.71% | 42.65% | 4.41% | 13.24% | 68 |
| PaperQA | 100% | 0% | 0% | 0% | 237 |

The drop in performance of the linear settings, *Vanilla RAG* and *No search*, show the advantage of an agent-based model that can call tools multiple times until it is satisfied with the final answer. Surprisingly enough, not using the LLM's latent knowledge (*no ask LLM*) also hurts performance, despite the benchmark being based on information after the cutoff date – we suggest that the useful latent knowledge we find LLMs to possess in Table 5 helps the agent use the best pieces of evidence. Lastly, due to the retrieval-first nature of LitQA, where only a single chunk from the original paper includes the answer, both *single citation* and *no summary LLM* settings perform comparably to standard PaperQA.

**Does PaperQA Hallucinate Citations?**  We assessed citation hallucinations from GPT-3.5, GPT-4, and Claude-2 using 52 questions from an earlier version of LitQA[3], where we ask the LLM to answer the question in long-form and provide citations. The citations were assessed manually to verify existence of the paper, the accuracy of the citation details, and the relevance of the cited paper to the provided answer. Scores were based on the total number of citations N, rather than number of answers. We show hallucination results in Table 4. Hallucinated citations are categorized as full hallucination, citation inaccuracy, or context irrelevance (the paper cited exists, but is irrelevant to the question). We tested numerous times, but no hallucinated citations were produced through PaperQA. Anecdotally, we have observed PaperQA citing a secondary source mentioned in a primary source which could lead to a hallucination since it has no access to the secondary source.

**Evaluation on QA Benchmarks** We evaluate PaperQA on standard QA multiple-choice datasets commonly used to assess LLMs. These benchmarks test for commonly known facts, which are most commonly not found in academic papers, but in textbooks. Here we measure how well PaperQA, which relies on information from papers, can perform on such questions. For all datasets, we evaluate on 100 randomly sampled questions (see Appendix E for details). We evaluate on the following datasets: **PubMedQA** (Jin et al., 2019) consists of yes/no/maybe questions that can be answered using a provided context. To mimic the LitQA setting, the context is not provided to the model. We call this PubMedQ$_{blind}$ to distinguish from PubMedQA. Furthermore, we omit questions that have a "maybe" ground-truth answer, as such questions may have evolved into definitive "yes" or "no" answers since the dataset was released in 2019. **MedQA** (Jin et al., 2020) consists of multiple-choice questions based on the United States Medical License Exams (USMLE) and covers multiple languages. We use the questions in English. **BioASQ** (Tsatsaronis et al., 2015) is a biomedical QA dataset. We only use the yes/no questions.

In Table 5 we compare PaperQA to GPT-4 and AuoGPT (we do not measure the performance of commercially available tools as they have no API and require manual evaluation). For all benchmarks and methods, we use zero-shot inference. While AutoGPT underperforms GPT-4, PaperQA outperforms GPT-4 in all datasets. We see that the biggest gap in performance, and the biggest improvement when equipping GPT-4 with PaperQA search is on PubMedQA$_{blind}$. In Appendix F we find that PaperQA matches the performance of GPT-4 that uses ground-truth context (which contains the correct answer by design), showing it is able to retrieve context information that is on par with the ground-truth one. We see a smaller gap between PaperQA and GPT-4 on MedQA, which we explain with the fact that MedQA is based on medical examinations and requires knowledge found

---

[3]The final revised dataset had fewer questions because some had multiple potential answers, so there is small discrepancy on the questions used here.

Table 5: **Evaluation of PaperQA on Standard QA Benchmarks.** We evaluate PaperQA on several multiple-choice QA datasets. $PubMedQA_{blind}$ is a version of the PubMedQA, where we obscure the context. GPT-4 + PaperQA is GPT-4 prompted to either answer or use PaperQA as an oracle to help answer the question. {PaperQA/AutoGPT} + Post reasoning involves feeding the output of PaperQA or AutoGPT to GPT-4, which is prompted to answer the question and use the output from the other systems if deemed useful. These two are similar to the *a priori* and *a posteriori* reasoning in Ren et al. (2023) respectively.

| Method | MedQA-USMLE | BioASQ | $PubMedQA_{blind}$ |
|---|---|---|---|
| GPT-4 | 67.0 | 84.0 | 57.9 |
| GPT-4 + PaperQA | 63.0 | 87.0 | **86.3** |
| AutoGPT | 54.0 | 73.0 | 56.8 |
| AutoGPT + Post reasoning | 56.0 | 75.0 | 61.3 |
| PaperQA | 68.0 | 89.0 | **86.3** |
| PaperQA + Post reasoning | **69.0** | **91.0** | 85.0 |

in textbooks, instead of academic papers. Finally, we make the observation that GPT-4, on average, performs better with *posteriori* reasoning when using PaperQA.

## 6 LIMITATIONS

PaperQA leverages fact, processes and concepts from underlying research papers; transforming that information into human and machine interpretable context. We have an underlying assumption from this that the information in the underlying papers is correct, which may not hold. Although we give some "signals" to the model like the journal name and citation count, these are not faithful indicators of quality.

Our models and benchmarks are affected by the changing nature of science and the availability of scientific literature: some of the questions in LitQA may have new correct answers or become invalid over time. Users of this benchmark should therefore limit papers used to answer the questions to be cutoff at September 15, 2023.

While recent work has been conducted on prompt optimization (Deng et al., 2022; Yang et al., 2023a), the complex setting of multiple agents with individual prompts is unsolved. Specifically, the task becomes a non-trivial, bi-level optimization problem. Consequently, discerning the impact of manual prompt adjustments becomes difficult. Thus, it is unlikely our prompts are optimal and it is difficult to assess which pieces of the prompts are necessary.

## 7 CONCLUSION

We introduced PaperQA, a Retrieval-Augmented Generative (RAG) agent that can answer scientific questions better than other LLMs and commercial products. We found PaperQA to be more cost-efficient than humans, while still retaining its accuracy on par with human researchers. We measured the hallucination rate of citations for recent LLMs to be between 40-60%, whereas we were not able to find a single hallucinated citation in PaperQA's responses. We also introduced LitQA – a benchmark of 50 questions that require retrieving information from full-text scientific papers. The PaperQA system works independently of the underlying model, with various combinations of Claude-2, GPT-3.5, and GPT-4 providing strong results on LitQA. The most important attributes of PaperQA are its ability to dynamically use RAG tools, retrieve full-text papers, and iterate on the answer through the agent's decision-making. We hope this open-source implementation of a scientific question-answering system illuminates the design of future RAG agents and tools that reduce hallucinations in LLMs. With such advancements, we believe scientific research will be carried at a fraction of the cost and a multiple of the speed, spurring faster innovation within the natural sciences. By augmenting researchers with a literature review tool, we aim to minimise the time spend searching literature, and rather maximize the amount of productive hours spend carrying out deep thinking for scientific research.

## REPRODUCIBILITY STATEMENT

Discussions on the inherent limitations and challenges faced, along with the mitigations applied to maintain the robustness and reliability of the models, are available in section 6. For comprehensive understanding and replicability of our approach, we recommend a thorough review of the mentioned section, coupled with the supplemental materials provided.

## ETHICS STATEMENT

Dual-use (applying technology for both beneficial and detrimental purposes) is an ongoing concern as more powerful models are developed Urbina et al. (2022). We performed rigorous red-teaming questions and did not observe risks for harm that are significantly elevated compared to direct use of the language models used for summarization. Similarly, the LitQA dataset and responses pose no risk for harm.

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

# Appendix

## A   PAPERQA IMPLEMENTATION DETAILS

### A.1   VERSIONS

The underlying model versions used are `GPT-3.5-turbo-0613` and `GPT-4-0613`. We used LangChain `v0.0.303` and OpenAI-python `v0.28.1`.

### A.2   SYSTEM PROMPT

The system prompt for the LLMs (`ask LLM`, `summary LLM`, and `answer LLM`) is given below:

```
Answer in an direct and concise tone, I am in a hurry. Your audience is an expert, so be
highly specific. If there are ambiguous terms or acronyms, first define them.
```

The system prompt of the agent is

```
You are a helpful AI assistant.
```

### A.3   TOOL DESCRIPTIONS

`search` description:

```
Search for papers to increase the paper count. Input should be a string of keywords. Use
this format: [keyword search], [start year]-[end year]. You may include years as the last
word in the query, e.g. 'machine learning 2020' or 'machine learning 2010-2020'. The
current year is get_year().
```

`gather_evidence` description:

```
Give a specific question to get evidence for it. This will increase evidence and relevant
paper counts.
```

`generate_answer` description:

```
Ask a model to propose an answer using evidence from papers. The input is the question to
be answered. The tool may fail, indicating that better or different evidence should be
found.
```

### A.4   ASK LLM PROMPT

To capture latent knowledge in LLMs, we use the following prompt:

```
We are collecting background information for the question/task below.
Provide a brief summary of information you know (about 50 words) that could help answer
the question - do not answer it directly and ignore formatting instructions. It is ok to
not answer, if there is nothing to contribute.

Question: question
```

## B   AUTOGPT IMPLEMENTATION DETAILS

We used LangChain's AutoGPT implementation from LangChain experimental 0.0.8 version. As AutoGPT can run indefinitely, we added a stopping condition after 10 searches. For LitQA, it is asked to answer the question based on all previous information; for other datasets, we restart it and make sure it chooses one of the options, so that we have an answer to all questions. The agent LLM used was GPT-4-0314. It has access to 3 tools: Google search, write to file, and read from file.

## C   PAPER RETRIEVAL EVALUATION

### C.1   ABSTRACT RETRIEVAL METRIC

As part of the development of PaperQA, we evaluate the efficacy of paper searching using different search API engines. In order to do that, we create a new metric composed of 500 questions. The goal is to create questions that are likely to have only a small number of papers that would include an answer. The exact methodology is:

1. Search PubMed with 20,000 unique tuples of five keywords within the field of medicine, biology and artificial intelligence.
2. Only keep the search results with a single paper that is not a review (where a review is assumed to have more than 20 authors).
3. Instruct GPT-4 to generate a question that could be answered by the abstract only.

The question generation follows a 5-shot prompting technique, the system prompt being:

```
You are a helpful assistant helping completing the following task. The goal is to create a
question that could be answered by the paper with these abstracts. Be creative and think
of a question that a scientist would ask without knowing the paper he is looking for.
```

A concatenated example of such abstract-question pair (red being the LLM completion) follows:

```
Title:
DeepDTA: deep drug-target binding affinity prediction

Abstract:
Motivation: The identification of novel drug-target (DT) interactions is a substantial
part of the drug discovery process. [...] One novel approach used in this work is the
modeling of protein sequences and compound 1D representations with convolutional neural
networks (CNNs).
Results: The results show that the proposed deep learning based model that uses the 1D
representations of targets and drugs is an effective approach for drug target binding
affinity prediction. [...]

Question:
Are there any models that use 1D representations of targets and drugs for drug target
binding affinity prediction?
```

This dataset is then used to search for the top-10 papers using 20 keywords that were generated using the following prompt:

```
We want to answer the following question: question
Provide num_keywords unique keyword searches (one search per line) and year ranges that
will find papers to help answer the question. Do not use boolean operators. "Make sure not
to repeat searches without changing the keywords or year ranges. Make some searches broad
and some narrow. The current year is get_year().
Use this format: 'X. [keywords], [start year]-[end year]'
For example, a list of 3 keywords would be formatted as:
1. 'keyword1, keyword2, 2020-2021
2. 'keyword3, keyword4, keyword5, 2020-2021
3. 'keyword1, keyword2, 2020-2021'
```

We thus evaluate recall, i.e. finding the original paper. Figure 2 shows the cumulative recall curve, where Google Scholar and Semantic Scholar show outstanding ability to retrieve the original paper. Although CORE API (Knoth et al., 2023) does not perform on par with the other two, we include it here as their main contribution to the field of scientific literature is their standardisation of open-access article repositories.

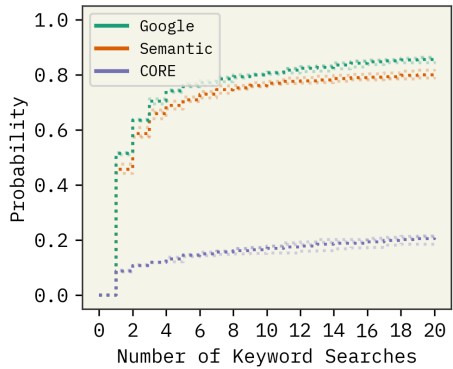

Table 6: **Retrieval AUC (Abstract).** We evaluate keyword generation and search between two LLMs and three search engines. The metric displayed is the normalized area under the curve from Figure 2 on the left.

| Search | LLM | Found |
|---|---|---|
| CORE | Claude-2 | 0.13 |
| | GPT-4 | 0.13 |
| Semantic | Claude-2 | 0.64 |
| | GPT-4 | 0.72 |
| Google | Claude-2 | 0.68 |
| | GPT-4 | **0.76** |

Figure 2: **Retrieval Probability (Abstract).** Cumulative probability distributions of finding, accessing and parsing the original paper from the described PubMed-based dataset by running API searches using GPT-4 for keyword generation. Uncertainty is shown by lighter shadows.

### C.2 Full-Text Retrieval Metric

As the abstract retrieval from Appendix C.1 does not sufficiently cover all the use cases of PaperQA, we also use an earlier version of LitQA to evaluate our search pipeline, but now considering questions that are synthesized from articles' bodies and not abstracts. Moreover, we examine the ability of our pipeline to find, access and parse the original papers of these questions.

Figure 3 and Table 7 shows superior performance of Google Scholar over Semantic Scholar. We believe this is thanks to Google Scholar's ability to search through the text of articles, whereas Semantic Scholar is limited to titles, authors and abstracts only. Effectively, we are able to parse almost all the papers we have access to. Notice the marginally better performance of GPT-4, which is likely due to it better following the instructions from the prompt. We expect that some prompt optimization might bring the performance of the two LLMs closer together. Lastly, note that since running this evaluation, we have improved our access to papers with Google Scholar's open-access links, so it is likely the final performance of parsed papers would be even better.

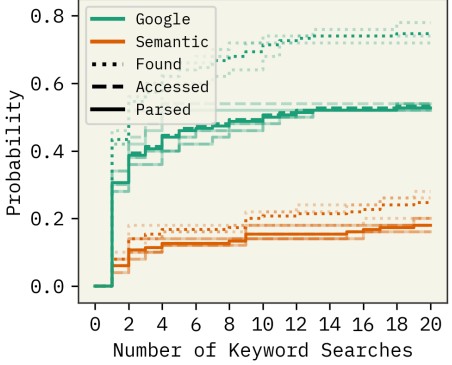

Figure 3: **Retrieval Probability (Full-Text).** Cumulative probability distributions of finding, accessing and parsing the original paper from LitQA by running API searches using GPT-4 for keyword generation.

Table 7: **Retrieval AUC (Full-Text).** We evaluate keyword generation and search between two LLMs and two search engines. We also evaluate our pipeline for accessing papers and parsing their PDFs. The metric displayed is the normalized area under the curve from Figure 3 on the left.

| Search | LLM | AUC of Figure 3 | | |
| --- | --- | --- | --- | --- |
| | | Found | Accessed | Parsed |
| Semantic | Claude-2 | 0.12 | 0.07 | 0.07 |
| | GPT-4 | 0.19 | 0.14 | 0.14 |
| Google | Claude-2 | 0.57 | 0.28 | 0.28 |
| | GPT-4 | **0.66** | **0.47** | **0.47** |

## D    HALLUCINATION DATASET

To evaluate hallucinations, the following prompt was given to each model:

```
Answer the question below, with citations to primary sources that help answer the
question. Cite the sources using format - (Foo et al., 2010) - note that the whole
citation is always in parantheses.
```

## E    EVALUATIONS ON STANDARD QA BENCHMARKS

For each dataset, we provide the questions ids, together with the formatted question and options list provided in the prompt. Please refer to the Supplementary Material.

## F    QUALITY OF DISCOVERED EVIDENCE

To evaluate the quality of the discovered evidence, we compare the PaperQA discovered evidence to the ground-truth evidence provided in PubMedQA, which is sufficient to answer the questions correctly by design. In Table 8 we show that the PaperQA discovered evidence is competitive to the ground-truth context. Furthermore, PaperQA finds complementary information to the one provided in the ground-truth context, further improving results when both are provided.

| Context | | PubMedQA |
| --- | --- | --- |
| Ground Truth | PaperQA | |
| ✗ | ✗ | 57.9 |
| ✓ | ✗ | 85.2 |
| ✗ | ✓ | 86.3 |
| ✓ | ✓ | 90.1 |

Table 8: **The quality of PaperQA-discovered context.** We compare providing the ground truth context in PubMedQA to the answer generated by PaperQA.

## G    IMPACT OF PARAMETRIC KNOWLEDGE

In order to test whether including parametric knowledge in the gathered context is useful, we prompt PaperQA:

```
Write an answer (about 200 words, but can be longer if necessary) for the question below
based on the provided context. If the context provides insufficient information and the
question cannot be directly answered, reply "I cannot answer". For each part of your
answer, indicate which sources most support it via valid citation markers at the end of
sentences, like (Example2012).
```

Followed by pre-gathered context with relevance scores, some extra background information, and the question. The extra background information represents the LLMs' parametric knowledge.

### G.1    CONTRADICTING INFORMATION

In the first example, we ask *Are COVID-19 vaccines effective?*. The goal of this experiment is to evaluate the inclusion of background information that contradicts gathered context. Thus, while the provided context supports the efficacy of covid vaccines, we provide the model with the following background information: "COVID-19 vaccines are known to be ineffective."

With the contradicting background information, PaperQA responds that it cannot answer. However, if the extra, contradicting information is excluded, the model provides an answer:

```
Yes, COVID-19 vaccines are effective. The BNT162b2 and ChAdOx1 nCoV-19 vaccines have shown
effectiveness against the delta variant, with the second[...]
...
Therefore, while the vaccines are effective, their effectiveness can vary based on the
specific vaccine, the variant of the virus, and the time elapsed since vaccination.
```

When the background information (or parametric knowledge) contradicts context, PaperQA opts to not answer, indicating its recognition of a contradiction with established scientific understanding. This example shows the model's capability to discern and respond appropriately to accurate information, while avoiding potentially misleading or incorrect assertions.

### G.2    ABSENCE OF KEY INFORMATION

As a second example, we ask PaperQA *What temperature does water turn to plasma?* and provide it with context that is relevant but insufficient to answer the question. Then, we provide the model with the following background information:

"Water turns into plasma under extreme conditions of temperature and pressure. Plasma is the fourth state of matter, beyond solid, liquid, and gas. It's a super-heated state where atoms are ionized, meaning electrons are stripped from atoms. This typically occurs at temperatures exceeding 20,000 Kelvin."

In this example, the model answers:

```
Water transitions into the plasma state under extreme conditions of temperature and
pressure, typically at temperatures exceeding 20,000 Kelvin.
```

Excluding the background information results in PaperQA responding that it is unable to answer.

Thus, the role of parametric knowledge is crucial in shaping the model's responses: when it contradicts the context, the model may refrain from answering due to the conflict, whereas supportive parametric knowledge enables the model to provide detailed and informed responses when context is lacking.

