# OpenReview forum: "PaperQA: Retrieval-Augmented Generative Agent for Scientific Research"
_ICLR.cc/2024/Conference — Submitted to ICLR 2024_

### Official Review · Reviewer_vdwn · 2023-10-23

**Soundness:** 2 fair
**Presentation:** 3 good
**Contribution:** 2 fair
**Rating:** 6
**Confidence:** 4

**Summary:**

The authors propose an agent-based scientific multiple choice QA system, mostly powered by LLMs and search APIs. They also contribute LitQA - 50 multiple choice questions written by experts.

**Strengths:**

I have read the author responses and have increased the score by 1 point.
-------------------------------------------------------

Thanks to the authors for their hard work on this agent and dataset.

Strengths:
- Good performance on the proposed dataset
- Clear writing
- Extensive ablations

**Weaknesses:**

- Is it likely you overfit the entire PaperQA system? There are only 50 questions. Was PaperQA developed after the data was collected? Did you make important system choices based on outcomes on the 50 questions? I think you need a development and test set split and I worry that the 50 questions in LitQA are actually the development set. Table 5 suggests that your system's advantage over GPT + search is actually much smaller than suggested by the deltas in Table 2.
- Multiple-choice is not realistic. Scientists don't already know what they're trying to solve when they are doing research. I think the entire paper should be the "No MC options" row from Table 3. I realize that MC means evaluation is easier, but with only 50 questions, manual evaluation of the various systems in Table 2 is feasible. Please consider rerunning the entirety of table 2 with the "No MC options".
- How *did* you evaluate the No MC options? Exact string overlap? Manually? I think it should be the latter.
- As I'm sure you know, Table 4 is hard to believe. Could it be because the MC options are included? What happens to the hallucination rate when the MC options are removed? I have done a lot of manual evaluation of many LLMs over the past year and have never seen a 0 hallucination rate. This needs to be more thoroughly understood as a part of this paper.

**Questions:**

- Will you be open sourcing your agent system?
- In table 3, what is the "Samples" column? Is this how many times you ran the entire end-to-end experiment? If so, can you include the standard deviations?
- Are Perplexity, Scite, and Elicit made to answer multiple-choice questions? I don't believe so. How do they compare in the "No MC options" setting?

---

> ### Author Response · Authors · 2023-11-23
>
> **LitQA - Dataset Size**
> > Is it likely you overfit the entire PaperQA system? There are only 50 questions. Was PaperQA developed after the data was collected? Did you make important system choices based on outcomes on the 50 questions? I think you need a development and test set split and I worry that the 50 questions in LitQA are actually the development set.
>
> This is a very reasonable concern. However, LitQA was only introduced as a performance metric after the PQA system was built. Thus, the performance on LitQA was not used for any hyperparameter optimization.
>
> **Table 5 - Clarification**
> > Table 5 suggests that your system's advantage over GPT + search is actually much smaller than suggested by the deltas in Table 2.
>
> To clarify, Table 2’s GPT row is an experiment without any search. We prompt GPT to answer a question from LitQA. In Table 5, we never use GPT-4 + search, there was a typo and we have fixed it in the new version of the paper. GPT-4 + PaperQA search referred to GPT-4 + PaperQA a priori reasoning, where GPT-4 is prompted to either answer the question directly, or use PaperQA as an oracle to help it answer the question.
>
> **LitQA - Multiple Choices**
> > Multiple-choice is not realistic. Scientists don't already know what they're trying to solve when they are doing research. I think the entire paper should be the "No MC options" row from Table 3. I realize that MC means evaluation is easier, but with only 50 questions, manual evaluation of the various systems in Table 2 is feasible. Please consider rerunning the entirety of table 2 with the "No MC options".
>
> Thank you for your feedback on our use of multiple-choice questions. We chose this approach for its ability to provide standardized, objective comparisons across systems, and to manage the practical constraints of our study's scale.
>
> In addition, while multiple choice questions may seem simplistic compared to the open-ended nature of scientific research, they are designed to encapsulate key elements of the problem-solving process. Each option represents a plausible hypothesis or outcome, mirroring the decision-making process scientists engage in when evaluating multiple possibilities.
>
> Nonetheless, we acknowledge the limitations of this method and will consider incorporating more open-ended questions in future research to address these concerns. Poor performance in Table 3’s ablation with No MC options - as you highlight - is a clear motivation to investigate new ways of evaluating open-ended QA settings.
>
> **LitQA - Evaluation**
> > How did you evaluate the No MC options? Exact string overlap? Manually? I think it should be the latter.
>
> We used GPT-4 to compare provided answer against reference answer. We manually checked one 50 question run to ensure it agreed with experts and validate this method.
>
> **Hallucination**
> > As I'm sure you know, Table 4 is hard to believe. Could it be because the MC options are included? What happens to the hallucination rate when the MC options are removed? I have done a lot of manual evaluation of many LLMs over the past year and have never seen a 0 hallucination rate. This needs to be more thoroughly understood as a part of this paper.
>
> Thank you for this comment. We have revised the manuscript to reflect that the dataset on which the hallucination evaluation was conducted was long-form, with no multiple choice options given. The 0% hallucination rate is largely influenced by the RAG step.
>
> **Open-Sourcing**
> > Will you be open sourcing your agent system?
>
> Yes, the original PaperQA system without an agent is already open-sourced on one of the author’s GitHub, where the agent version will be open-sourced thereafter.
>
> **Table 2 Clarification**
> > In table 3, what is the "Samples" column? Is this how many times you ran the entire end-to-end experiment? If so, can you include the standard deviations?
>
> That is indeed correct - “Samples” refers to the number of times the experiment with the agent was run end-to-end. We keep averages for “Response” columns, but will include standard deviations for the “Score” in the camera-ready paper.
>
> **Industrial Competitors**
> > Are Perplexity, Scite, and Elicit made to answer multiple-choice questions? I don't believe so. How do they compare in the "No MC options" setting?
>
> When we performed these experiments, we prompted PaperQA and all other models identically, meaning that each had the question, along with the multiple choice options.
>
> The mentioned models are not necessarily designed to answer multiple choice questions, but the multiple choice options serve to provide the model with additional context, rather than limitation in this case. Often the competitor models would answer a long-form answer that matched the context of a provided answer, and of course, in this case, the manual evaluation shows correct.

---

### Official Review · Reviewer_CSpW · 2023-10-28

**Soundness:** 2 fair
**Presentation:** 3 good
**Contribution:** 3 good
**Rating:** 5
**Confidence:** 3

**Summary:**

This paper describes PaperQA, an agent that answers questions about scientific literature according to the search results. The agent is composed of three tools: search, gather evidence, and answer the question. It can find and parse relevant full-text research papers, identify specific sections in the paper that help answer the question, summarize those sections with the context of the question (called evidence), and then generate an answer based on the evidence. Compared to a standard retrieval-augmented generative (RAG) agent, PaperQA decomposes parts of a RAG and provides them as tools to an agent. It can adjust the input to paper searches, gather evidence with different phrases, and assess if an answer is complete.

**Strengths:**

1. PaperQA decomposes parts of a RAG and provides them as tools to an agent, and it can adjust the input to paper searches, gather evidence with different phrases, and assess if an answer is complete.
2. PaperQA makes use of a priori and a posteriori prompting, tapping into the latent knowledge in LLMs.
3. PaperQA outperforms all models tested and commercial tools, and is comparable to human experts on LitQA on performance and time.

**Weaknesses:**

1. The paper has some innovation, but it still feels limited. Firstly, the dynamic use of the three tools is quite similar to the ReAct framework, all of which are dynamically autonomous in determining whether to retrieve them again. Secondly, the number of benchmarks constructed is relatively small, with only 50 questions and a multiple-choice format. Existing research has shown that the form of multiple choice questions has limitations in evaluating model performance, and the model is more often used to generate longer texts. Therefore, there is a significant gap between the form of multiple choice questions and practical applications.
2. In the experiment, there is a lack of comparison with some advanced agent frameworks, which often consider the dynamic nature of intermediate steps. Therefore, it is necessary to increase the comparison with these frameworks, such as ReAct and Reflexion. The main experiment is conducted on multiple choice questions, which has limitations because hallucinations typically occur when the generated text is long. During the hallucination evaluation experiment, some details were not clearly written, such as whether other LLMs used search tools.

**Questions:**

N/A

---

> ### Author Response · Authors · 2023-11-23
>
> **Other Agent Frameworks**
> > Firstly, the dynamic use of the three tools is quite similar to the ReAct framework, all of which are dynamically autonomous in determining whether to retrieve them again.
> In the experiment, there is a lack of comparison with some advanced agent frameworks, which often consider the dynamic nature of intermediate steps. Therefore, it is necessary to increase the comparison with these frameworks, such as ReAct and Reflexion.
>
> We thank the reviewer for this comment. We do not necessarily need to compare with other agent frameworks, because we are not proposing a new agent framework. We rather propose a system built on top of an arbitrary agent framework, able to think and use tools.
>
> More specifically, we use the OpenAIFunctionsAgent from LangChain. Nevertheless, we also tested the ReAct agent without any major difference in performance. If required, we are happy to provide those results as an ablation in the final version of the paper.
>
> **LitQA**
> > Secondly, the number of benchmarks constructed is relatively small, with only 50 questions and a multiple-choice format. Existing research has shown that the form of multiple choice questions has limitations in evaluating model performance, and the model is more often used to generate longer texts. Therefore, there is a significant gap between the form of multiple choice questions and practical applications.
>
> The 50 question LitQA dataset was laboriously created to evaluate PaperQA  on a small subsection of papers that the underlying LLMs had not seen in training data. We also evaluated PaperQA on several existing evaluation benchmarks that are multiple choice.
>
> While we agree that multiple choice evaluations have their limitations, the multiple choice format allows us to properly evaluate performance across multiple models, as well as humans, without any bias. Thus, this evaluation metric has benefits that outweigh the mentioned limitations.
>
> **Hallucination**
> > The main experiment is conducted on multiple choice questions, which has limitations because hallucinations typically occur when the generated text is long. During the hallucination evaluation experiment, some details were not clearly written, such as whether other LLMs used search tools.
>
> The hallucination evaluation was carried out in long-form, requiring all models to give citations to support all claims. Then, every citation (with respect to the corresponding claim, when applicable) was evaluated manually to determine whether it was hallucinated or not.
>
> The hallucination experiment was conducted to determine hallucinated citations, showing the power of PaperQA to produce claims, backed by correct and valid citations, compared to simple LLMs (no tools).

---

### Official Review · Reviewer_afKA · 2023-11-01

**Soundness:** 2 fair
**Presentation:** 2 fair
**Contribution:** 2 fair
**Rating:** 3
**Confidence:** 5

**Summary:**

This paper presents PaperQA, a tool developed with retrieval-augmented generation (RAG) technique to answer science questions. They also proposed LitQA, new benchmark to assess the performance of RAG models.

**Strengths:**

* Authors introduce new components to the standard RAG pipeline (e.g., search, map-reduce the summary, repeat for more evidence)
* Adaptive and modular framework and an implementation with open source libraries.

**Weaknesses:**

* This paper appeared to be more product or application specific than focused on the underlying research problems. Unfortunately, no research problem was mentioned in the text.
* Ask LLM prompt assess the parametric knowledge, which is feed to the evidence contexts. Authors found this knowledge is helpful. But I do not agree with the reasons provided. For example, what would happen if there are knowledge conflicts raised with the parametric knowledge and retrieved knowledge?
> “Surprisingly enough, not using the LLM’s latent knowledge (no ask LLM) also hurts performance, despite the benchmark being based on information after the cutoff date – we suggest that the useful latent knowledge we find LLMs to possess in Table 5 helps the agent use the best pieces of evidence.”
* I don’t think the following claim is true. Having a low rate of incorrect answers does not suggest that the model is certain, in fact, one needs to measure the uncertainty in the generated answers to make such a claim.
“Furthermore, we see the lowest rate of incorrectly answered questions out of all tools, which rivals that of humans. This highlight’s PaperQA’s ability to be certain about its answers.”
* This is a bad analogy, I don’t think that human judgmental time should need to correlate with the time taken to complete OpenAI API calls.
> “It took PaperQA on average about 2.4 hours to answer all questions, which is also on par with humans who were given 2.5 hours.”

**Questions:**

* What kind of reasoning required in this case? I can only find the task is to measure the relevance of the query to the retrieved passages. It is intriguing why authors opt out the model to explain the score.
> “the LLM’s ability to reason over text and provide numerical relevance scores for each chunk of text to the query question.”
> “At the end of your response, provide a score from 1-10 on a newline indicating relevance to question. Do not explain your score”
* How authors reliably make the claim of the GPT4 cut off date? Which GPT4 version used in the study?
> “We take special care to only collect questions from papers published after the GPT-3.5/4 cutoff date in September 2021”
* Is it expected that biomedical researchers cannot answer these question without internet? Or is this setup introduced to mimic the RAG styled QA by asking only to answer given what they find on the internet?
> “We recruited five biomedical researchers with an undergraduate degree or higher to solve LitQA. They were given access to the internet and given three minutes per question (2.5 hours in total) to answer all questions”
* This is unexpected, any reasons? Is the context length a factor to correlate with the summarization performance?
> “Interestingly, we observe that using GPT-4 as the summary LLM worsens overall performance.”

---

> ### Author Response · Authors · 2023-11-23
>
> **No Research Problem**
> > This paper appeared to be more product or application specific than focused on the underlying research problems. Unfortunately, no research problem was mentioned in the text.
>
> We appreciate this critique and agree that a clear problem statement should be defined. Thus, we have adjusted our manuscript to reflect this. We tackled the three following problem statements:
> 1. *Does the inclusion of a LLM-driven, non-linear RAG workflow aid in carrying out scientific question-answering?*
> In this question, we were particularly interested in PaperQA as an agent system, where the pieces of RAG could be performed modularly and iteratively as needed, without human interaction. We found in our ablation study that PaperQA, which was allowed to follow any ordering and number of tools, outperformed Vanilla RAG, which was required to follow a linear workflow.
> 2. *Are LLMs augmented with RAG-style tools superior on established benchmarks?*
> For this question, we evaluated our method, as well as other methods, on established benchmarks and found that PaperQA indeed does reach higher performance, suggesting the RAG-style tools do indeed boost performance.
> A sub-question to this is whether our full-text document and chunk summary methods boost performance, which we indeed confirm in our manuscript.
> 3. *Does hierarchical RAG with access to full-text scientific articles lead to robust answers without hallucinated references?*
> We evaluated this and compared to LLMs with no tools, and we found that indeed, while plain LLMs do hallucinate citations (which is expected), our method is robust.
>
> **Ask LLM**
> > Ask LLM prompt assess the parametric knowledge, which is feed to the evidence contexts. Authors found this knowledge is helpful. But I do not agree with the reasons provided. For example, what would happen if there are knowledge conflicts raised with the parametric knowledge and retrieved knowledge?
>
> This is an excellent point. We have added an example outlining this in the appendix (Appendix G).
>
> In summary, the role of background information is crucial in shaping the model's responses: when it contradicts the context, the model may refrain from answering due to the conflict, whereas accurate and supportive background information enables the model to provide detailed and informed responses.
>
> **Precision & Certainty**
> > I don’t think the following claim is true. Having a low rate of incorrect answers does not suggest that the model is certain, in fact, one needs to measure the uncertainty in the generated answers to make such a claim. “Furthermore, we see the lowest rate of incorrectly answered questions out of all tools, which rivals that of humans. This highlight’s PaperQA’s ability to be certain about its answers.”
>
> We agree that our wording was imprecise. We do not mean to say that PQA is more certain, but rather that it is better calibrated to express uncertainty when it is uncertain. In comparison, GPT-4’s RLHF training results in poor calibration in this respect, as it tries to give full answers to please the asker. This means that it is more confident than it should be in some cases. Our system, with suitable prompt engineering, alleviates this. We have changed the section highlighted to the following:
>
> “Furthermore, we see the lowest rate of incorrectly answered questions out of all tools, which rivals that of humans. This emphasizes that PaperQA is better calibrated to express uncertainty when it actually is uncertain.”
>
> **Human vs. PQA Time**
> > This is a bad analogy, I don’t think that human judgmental time should need to correlate with the time taken to complete OpenAI API calls.
>
> We understand the reviewers concern, but here we do not claim that these two correlated, but that rather PQA does not necessarily take longer and thus can be competitive with humans on performance, as well as time.

---

> ### Author Response · Authors · 2023-11-23
>
> **Summary LLM**
> > What kind of reasoning required in this case? I can only find the task is to measure the relevance of the query to the retrieved passages. It is intriguing why authors opt out the model to explain the score.
> > “the LLM’s ability to reason over text and provide numerical relevance scores for each chunk of text to the query question.” “At the end of your response, provide a score from 1-10 on a newline indicating relevance to question. Do not explain your score”
>
> Due to context length limits, we provided 8 contexts for the final answer. To populate these 8, we performed the summary over 20 potential source documents. This is because some will likely be irrelevant. However, to downsample from 20 to 8, we need a way to rank the documents to choose the best 8. One method would be to have multiple LLM calls to provide relative rankings while avoiding context limits. However, our approach was to have the summary LLM to provide an estimated numerical ranking that aids in ranking the summaries.  We did not explore the calibration or agreement with experts on this number - the scores are a quick concurrent method used to downselect the summaries to 8.
>
> **GPT Version**
> > How authors reliably make the claim of the GPT4 cut off date? Which GPT4 version used in the study? “We take special care to only collect questions from papers published after the GPT-3.5/4 cutoff date in September 2021”
>
> The GPT-3.5 version used was gpt-3.5-turbo-0613 and GPT-4 version was gpt-4-0613. We have added technical details for all accessed and utilized models in the SI.
>
> In terms of the cutoff date, we rely on OpenAI’s documentation where they specifically mention all training data originates from before September 2021. Moreover, the poor results of Claude-2 and GPT-4 in Table 2 show that most of the papers were unlikely to be in the training dataset. We still however see that both models appear to have some inner knowledge to help them be biased towards correct answers on average, which is expected.
>
> **LitQA**
> > Is it expected that biomedical researchers cannot answer these question without internet? Or is this setup introduced to mimic the RAG styled QA by asking only to answer given what they find on the internet?
>
> As the questions require a detailed and specific retrieval of very recent knowledge from the literature, we do not expect any biomedical researcher to be able to answer these without the use of the internet. The answers for the LitQA questions are also not answered in the abstract of the papers, by design, requiring the researcher to dig into the paper for the information. LitQA is thus used to evaluate the process of scientific literature retrieval itself, comparing between RAG and human researchers.
>
> **Summary LLM**
> > This is unexpected, any reasons? Is the context length a factor to correlate with the summarization performance? “Interestingly, we observe that using GPT-4 as the summary LLM worsens overall performance.”
>
> Thank you for pointing this claim out. As it is unsupported in our manuscript and not central to the purpose of the manuscript, we have removed this claim altogether.

---

### Official Review · Reviewer_1BNP · 2023-11-02

**Soundness:** 2 fair
**Presentation:** 2 fair
**Contribution:** 2 fair
**Rating:** 3
**Confidence:** 4

**Summary:**

The paper presents PaperQA, a Retrieval-Augmented Generation (RAG) agent developed to enhance question-answering in the scientific domain by mitigating issues of hallucinations and uninterpretability associated with Large Language Models (LLMs). Unlike other LLMs, PaperQA searches and retrieves information from full-text scientific papers to generate more accurate and interpretable responses. The authors showcase PaperQA's performance over existing LLMs on science QA benchmarks and introduce a new benchmark called LitQA, designed to simulate the complex task of human literature research. PaperQA is said to perform on par with expert human researchers when evaluated against the LitQA benchmark.

**Strengths:**

The concept introduced in the paper is promising, as it aims to develop a framework for retrieving literature to facilitate the answering of questions within scientific texts. The authors propose a novel approach that breaks down the QA task into three primary components: identifying relevant papers from online databases, extracting text from these papers, and synthesizing the information into a coherent final answer.

The paper introduces a new dataset, LitQA, which necessitates the retrieval and synthesis of information from full-text scientific papers. This is a notable effort to replicate the complexity of real-world scientific inquiry.

The study compares the proposed PaperQA system against multiple baselines. The results indicate that PaperQA outperforms these baselines and is on par with human experts.

**Weaknesses:**

Lack of Novelty:

The methodology presented in this paper follows the established pipeline of retrieval, reading, and answering, which has been extensively explored in prior literature. The paper does not adequately differentiate the proposed model from existing work in the field. For this approach to be considered a substantial contribution, it would require either a novel application of these methods or significant improvements over existing models, neither of which are sufficiently demonstrated in the current paper.

Insufficient Dataset Size:

The introduction of the LitQA dataset is an interesting addition; however, with only 50 examples, it is far too limited to serve as a robust benchmark for this area of research. Benchmarks require extensive and diverse examples to evaluate the generalizability and effectiveness of the proposed approach and to provide a reliable comparison with other baselines. The dataset, as it stands, does not meet these criteria.

Technical Feasibility and Lack of Detail:

There are concerns regarding the technical feasibility of some experimental settings described. Specifically, the instruction for the summary LLM to score relevance from 1 to 10 is not grounded in a clearly defined metric, raising questions about the model's capacity to interpret and apply these scores accurately.

Moreover, the paper omits crucial details necessary for the reproducibility of the results and the clarity of the methods used. For instance, the base LLM utilized for PaperQA is not specified, leaving a gap in understanding the foundation upon which the system is built. Similarly, the engines powering GPT-3.5 and GPT-4 are not clearly defined. The configurations and model setups for the tools Elicit, Scite_, Perplexity, and Perplexity (Co-pilot) are insufficiently detailed. This lack of clarity hinders the assessment of the methods and the comparison of the results.

**Questions:**

Missing related work:

- Are You Smarter Than a Sixth Grader? Textbook Question Answering for Multimodal Machine Comprehension
- Learn to Explain: Multimodal Reasoning via Thought Chains for Science Question Answering

Typos:
- “This implementation decision is explained” -> “The implementation details are explained”

---

> ### Author Response · Authors · 2023-11-23
>
> **Lack of Novelty**
> > The methodology presented in this paper follows the established pipeline of retrieval, reading, and answering, which has been extensively explored in prior literature. The paper does not adequately differentiate the proposed model from existing work in the field. For this approach to be considered a substantial contribution, it would require either a novel application of these methods or significant improvements over existing models, neither of which are sufficiently demonstrated in the current paper.
>
> Thank you for this comment. We have added a few details to the paper to address this comment directly, highlighting the novelty and value of PaperQA more clearly. PaperQA exceeds existing LLM-only models like GPT-4 by 29 points on PubMedQA, a closed-book benchhmark, and PaperQA beats all existing models/agents/commercial projects on our smaller dataset. The dataset had to be small because it required dozens of experiments - we had to manually enter and evaluate questions (e.g., using Perplexity.AI), and we had to use large amounts of expert time to generate questions, where one needs to check “has anyone reported X in the literature”  across the scientific literature.  We believe to have reported a model better than all existing LLMs and agent architectures on the class of doing scientific research, which we hope would be significant enough for acceptance.
>
> Regarding what is novel about PaperQA: PaperQA is novel in three ways.
>
> 1. First, we break the steps of RAG into tools that are accessible by the agent, allowing the agent to iterate and repeat steps as needed, in any order, in order to fully answer the question.
> 2. In addition, PaperQA has access to the majority of all available scientific literature, going beyond just PubMed, or just Arxiv, which has been done before.
> 3. Last but not least, we aim to show that using LLMs for difficult scientific questions is insufficient and that fine-tuning on specific scientific tasks may be unnecessary when compared to augmenting LLMs with the appropriate tools.
>
> **Insufficient Dataset Size**
> > The introduction of the LitQA dataset is an interesting addition; however, with only 50 examples, it is far too limited to serve as a robust benchmark for this area of research. Benchmarks require extensive and diverse examples to evaluate the generalizability and effectiveness of the proposed approach and to provide a reliable comparison with other baselines. The dataset, as it stands, does not meet these criteria.
>
> We agree that 50 questions is a limited dataset. Those 50 questions were evaluated in multiple experiments with various commercial products and LLMs, requiring a great deal of human time to enter questions.
>
> The focus of this paper was primarily the development and use of PaperQA. The LitQA dataset was an additional set that also was explicitly out of pre-training data of GPT-4/3.5. We believe that beating GPT-4 by 30 points on PubMedQA is a good assessment on a larger dataset.
>
> **Summary LLM**
> > There are concerns regarding the technical feasibility of some experimental settings described. Specifically, the instruction for the summary LLM to score relevance from 1 to 10 is not grounded in a clearly defined metric, raising questions about the model's capacity to interpret and apply these scores accurately.
>
> We appreciate this concern that the ability to assign this relevance only arises from the inherent capability of the base LLM, GPT-4. The relevance scoring metric is used to rank chunks, and the aim of this paper is to show that even with a general purpose model like GPT-4, the entire PQA workflow is able to find the relevant chunks and outperform other models on benchmarks.
>
> **Industrial Competitors**
> > The configurations and model setups for the tools Elicit, Scite_, Perplexity, and Perplexity (Co-pilot) are insufficiently detailed.
>
> We agree that the technical details of these are not disclosed, as most information regarding the model setups are proprietary. Thus, we have provided all information we have access to. In the spirit of transparency, we clarify that we prompted these models the same way that we prompted PaperQA and evaluated manually.
>
> **Reproducibility**
> > Moreover, the paper omits crucial details necessary for the reproducibility of the results and the clarity of the methods used. For instance, the base LLM utilized for PaperQA is not specified, leaving a gap in understanding the foundation upon which the system is built. Similarly, the engines powering GPT-3.5 and GPT-4 are not clearly defined.
>
> The GPT-3.5 version used was gpt-3.5-turbo-0613 and GPT-4 version was gpt-4-0613. These details have been added to the appendix (Appendix A), along with package version specifications.
>
> **Missing Related Work**
>
> Thank you for pointing us to these related works. We have added them in our manuscript.
>
> **Typos**
>
> Fixed.

---

### Meta-Review · Area_Chair_CTzJ · 2023-12-06

**Metareview:**

The reviewers have raised some significant concerns regarding the submission:

1. Limited Novelty: The proposed method appears to be a retrieval-augmented generation framework, a commonly used approach within our community. This raises questions about the novelty of the work.

2. Small Dataset: One of the claimed contributions of the paper is the introduction of a new benchmark, LitQA, which unfortunately contains only 50 examples. This small dataset size is a concern.

3. Weak Contribution: Given the issues highlighted in points 1 and 2, the remaining contributions of this work seem to be primarily centered around the evaluation aspect. There are several concerns raised in the reviews about the details of the evaluation that need to be addressed.

The author's responses have not adequately addressed concerns 1 and 2.

Furthermore, from the author responses, we see that the authors acknowledged that annotation of LitQA is time-consuming and the size of 50 is limited.. However, it is concerning that no efforts have been made to increase the dataset size since the paper's submission. This lack of improvement in dataset size reduces the confidence of both the reviewers and the AC in the potential for this work to make significant advancements in the camera-ready version. Accepting the paper as is may not guarantee substantial improvements in the final version.

**Justification For Why Not Higher Score:**

The author's responses have not adequately addressed the major concerns 1 and 2.

Furthermore, from the author responses, we see that the authors acknowledged that annotation of LitQA is time-consuming and the size of 50 is limited.. However, it is concerning that no efforts have been made to increase the dataset size since the paper's submission. This lack of improvement in dataset size reduces the confidence of both the reviewers and the AC in the potential for this work to make significant advancements in the camera-ready version. Accepting the paper as is may not guarantee substantial improvements in the final version.

**Justification For Why Not Lower Score:**

N/A

---

### Decision · Program_Chairs · 2024-01-16

Reject